# Electroplastic Effect during Tension and Bending in Duplex Stainless Steel

**DOI:** 10.3390/ma16114119

**Published:** 2023-05-31

**Authors:** Mikhail Pakhomov, Oleg Korolkov, Mirko Pigato, Claudio Gennari, Irene Calliari, Vladimir Stolyarov

**Affiliations:** 1Mechanical Engineering Research Institute of RAS, 101990 Moscow, Russia; 2Department of Industrial Engineering, University of Padova, 35131 Padova, Italy

**Keywords:** electroplastic effect, electroplastic deformation, tension, bending, current, heating, stainless steel

## Abstract

The deformation behavior of duplex stainless steel under tension and bending, accompanied by a pulsed current and when heated by an external source, is investigated. The stress–strain curves are compared at the same temperatures. The contribution to the decrease in flow stresses is greater when using a multi-pulse current at the same temperature, compared to external heating. This confirms the presence of an electroplastic effect. An increase in the strain rate by an order of magnitude reduces the contribution of the electroplastic effect from single pulses to the reduction in flow stresses by 20%. An increase in the strain rate by an order of magnitude reduces the contribution of the electroplastic effect from single pulses to the reduction in flow stresses by 20%. However, in the case of a multi-pulse current, the strain rate effect is not observed. Introducing a multi-pulse current during bending reduces the bending strength by a factor of two and the springback angle to 6.5.

## 1. Introduction

The electroplastic effect (EPE) has been extensively studied and documented in original articles [1,2] and reviews [3,4]. The EPE is characterized by a reduction in flow stresses and an increase in deformability during various metalworking processes such as tension [1], compression [5], bending [6], rolling [7], and drawing [8]. This phenomenon is of practical interest due to its potential to combine plastic deformation and current injection in metalworking, eliminating the need for intermediate and final furnace annealing. The scientific significance lies in understanding the physical mechanisms of the EPE, which extend beyond the traditional thermal effect and include pinch, spin, and magnetoelastic effects when a pulsed current passes through a conductor [3,9,10]. It is crucial to differentiate the potential contributions of each of these effects.

The EPE has been investigated in pure metals, alloys, and even ceramics [2]. Among various materials, stainless steels, particularly those used as structural materials in industries such as automotives and aviation, are of great interest. These steels not only require a favorable combination of high strength and satisfactory ductility but also a low springback. Studies have been conducted on low-carbon steels in different structural states, under various current modes and deformation patterns [11,12,13,14,15,16]. For example, the EPE in steels with a chemical composition similar to that of AISI 304L was studied during drawing with a pulsed current [11,12,13]. AISI 316L stainless steel in the martensitic state was investigated under tension with a constant current [13], and AISI 1010 steel was examined during rolling with a pulsed current [15]. Furthermore, electrotechnical silicon steel has been studied using indentation [16]. In metastable TRIP steel in the A + M state, researchers found that the EPE occurs during tension and leads to a greater softening than the thermal effect of a pulsed current [17]. The effect of the EPE in this case is more pronounced at higher current densities and strain levels. However, in metastable two-phase ferrite–martensite-steel DP980 AHSS, the flow stresses increase while the relative elongation decreases, indicating the absence of the EPE. This is attributed to the decomposition of martensite and the precipitation of carbide particles [17]. More recently, researchers have focused on studying the EPE in relatively stable ferrite–austenite steels, which exhibit an excellent combination of ductility, strength, and deformability. The effect of stacking-fault energy (SFE) on the manifestation of the EPE under tension with a direct current has been investigated [18].

An additional and unexplored feature of two-phase austenitic–ferritic (A + F) duplex stainless steel is the different sensitivity of its phases to an external magnetic field. Austenite is a paramagnetic phase, while ferrite is ferromagnetic [19,20]. Therefore, the deformation behavior of such steel under a pulsed current may differ from that of purely austenitic steels. The pulsed magnetic field induced by the current during plastic deformation can lead to additional pinch, skin, and magnetoplastic effects [21].

The strain rate is an important parameter for understanding and applying the EPE. Although there are limited data available in the literature on the EPE at different strain rates, the focus has primarily been on values greater than 10^−1^ s^−1^, disregarding the lower range of rates below 10^−2^ s^−1^, particularly for steels [22].

The influence of different current modes, regimes, and deformation schemes on the mechanical behavior and properties of duplex austenitic–ferritic stainless-steel UNS S32750 (DSS) is investigated in this work. It aims to explore the effects of various current parameters and deformation conditions on the material’s response.

## 2. Materials and Research Methods

For this study, a 2 mm thick sheet of UNS S32750 duplex stainless steel was chosen, supplied by the Italian division of Outokumpu S.r.l. The sheet was obtained through cold rolling. The chemical composition of the steel is presented in Table 1. To remove the work-hardened state resulting from cold rolling, an annealing treatment was conducted at 1020 °C for 10 min, followed by water quenching. Additionally, a solution treatment was performed at 1080 °C for 1 h to dissolve any potential secondary phases and restore the optimal volume percentage of ferrite and austenite to 50/50.The shape and dimensions of the specimens for tension and bending, prepared by the method of electric spark cutting along the rolling direction, are shown in Figure 1.

The microstructure was examined using a Leica DMRE (Germany) optical microscope, and fracture images were obtained using a scanning electron microscope—the Leica Cambridge Stereoscan LEO 440 (Germany).

Tension and bending tests were conducted using an IR-5081/20 (Ivanovo, Russia) horizontal tensile testing machine. The applied tensile (strain) rates were v (έ) = 0.6 (3 × 10^−4^ s^−1^), 6 (3 × 10^−3^ s^−1^), and 60 (3 × 10^−2^ s^−1^) mm/min. The bending speeds were set at 5 and 200 mm/min.

For the tension tests, the following modes were employed: (a)No current applied.(b)Single pulses with an amplitude current density of j = 500, 550, and 740 A/mm^2^ and pulse durations of τ = 250 and 1000 μs.(c)Multi-pulse current with densities of j = 15 and 45 A/mm^2^, pulse durations of τ = 100 and 900 μs, and a frequency of 1000 Hz.(d)Heating using a technical dryer to reach a temperature of 190 ℃, corresponding to the multi-pulse current mode with a density of j = 45 A/mm^2^ and a duration of τ = 100 μs.

The selection of current modes was based on the capabilities of the generator, the existing literature on duplex stainless steels, and the condition j > jcr [2]. The temperature of the sample was monitored using a Digital Thermometers UT320 Series and a chrome-alum thermocouple positioned at the center of the sample with an accuracy of ±2 °C. The current supply circuit for tension and bending is shown in Figure 2. Tension and bending tests were initiated after the chosen temperature stabilized.

In the bending process, the following modes were employed: (a) No current applied. (b) Multi-pulse current with a density of j = 20 and 30 A/mm^2^, a pulse duration of τ = 100 μs, and a frequency of 1000 Hz. (c) Heating using a technical dryer to reach a temperature of 100 ℃, corresponding to the multi-pulse current mode with a density of j = 20 A/mm^2^ and a pulse duration of τ = 100 μs.

The bending of the samples was conducted using specialized equipment with a distance of 36 mm between the supports. Fiberglass spacers were used to isolate the equipment from the test machine. The pulse generator current was applied to the ends of the sample, and the current modes were monitored using an oscilloscope.

To estimate the springback, the bending angles were compared in the highly loaded state after a 10 s hold and after the load was removed. Two samples were tested for each mode. In cases where there was a notable variation in the results, an additional test was performed.

## 3. Results

### 3.1. As-Received Microstructure

The microstructure of the as-received sample consists of fragmented austenitic grains dispersed within a ferritic matrix (Figure 3). The rolling direction is indicated by the double-pointed white arrow.

The microstructure of the as-received sample consists of fragmented austenitic grains dispersed within a ferritic matrix (Figure 3). The rolling direction is indicated by the double-pointed white arrow. There are no secondary phases observed at the grain boundaries of ferrite or at the phase boundaries between austenite and ferrite. Any potential secondary phases, if present, would appear as bright spots in backscattered electron images due to their higher content of high-atomic-number elements such as molybdenum. The microstructure is examined using a Leica DMRE optical microscope, and fracture images are obtained using a scanning electron microscope—the LEICA Cambridge Stereoscan LEO 440.

The fragmentation of austenitic grains is a result of the final pass during the rolling process, which is carried out at a low temperature. After the solution treatment, the austenite grains exhibit annealing twins due to their low stacking fault energy. Figure 3b shows an optical micrograph of the as-received sample along the three main directions: Transverse Direction (TD), Rolling Direction (RD), and Normal Direction (ND). The interphase space between austenite and ferrite grain centers is smaller along the normal direction compared to the rolling direction, attributable to the forming process.

### 3.2. Tension

Figure 4 shows the stress–strain curves of samples tested without current (curves 1, 2), with single-current pulses, and with different strain rates (curves 3, 4, 5). 

It is evident that in the tension test without current (curves 1 and 2), increasing the strain rate by two orders of magnitude results in a 10% decrease in elongation. Single-current pulses with varying densities, pulse durations, and frequencies (curves 3, 4, 5) lead to downward stress jumps in the plastic region, with amplitudes of up to Δσ = 15 MPa. Higher strain rates and lower pulse frequencies contribute to a reduction in the number of jumps and even their disappearance in the strain curves. This also leads to an increase in ultimate strength/yield stress and a decrease in elongation.

Interestingly, during the tension test with single pulses and the lowest strain rate in the elastic region, smaller-amplitude stress jumps of up to Δσ = 3 MPa are also observed (Figure 4, curve 5). However, as the tension rates exceed 3 × 10^−4^, the stress jumps in the elastic region decrease and eventually vanish.

Figure 5 illustrates the stress–strain curves of samples tested under different conditions: without current (curves 1, 2), with various modes of multi-pulse current (curves 6, 7, 8), and with the application of a technical dryer (curve 9).

When a multi-pulse current is applied, stress jumps become practically invisible. Increasing the density of the multi-pulse current leads to a decrease in flow stresses and elongation (Figure 5, curves 7, 8). On the other hand, heating with a technical dryer increases elongation and affects the position of curve 9, which is located above curve 8 for a multi-pulse current at the same temperature.

Table 2 presents the tensile conditions and mechanical properties of duplex stainless steel. Introducing current or applying heat with a technical dryer results in a decrease in strength and elongation, with the magnitude varying depending on the specific modes. On average, the strength decreases by 100–185 MPa and elongation decreases by 9–14%.

### 3.3. Microstructure 

Figure 6 displays the microstructures observed far away from the fracture surface of the samples tested under different conditions. The images correspond to the sample tested without current (A), with a single pulse at 740 A/mm^2^ (B), with a multi-pulse current at 45 A/mm^2^ (C), and with the application of a technical dryer (D).

A noticeable deformation and reorientation of both ferritic and austenitic grains can be observed in the vicinity of the fracture surface. This deformation and reorientation are a result of the triaxial stress state that occurs after reaching plastic instability following uniform elongation. However, when moving away from the fracture surfaces, the microstructure appears similar to that of the as-received sample, with only a slight elongation of the ferritic and austenitic grains due to the deformation process.

In Figure 7, the fractographs of a sample tested under different conditions are presented. Fractographs A and B correspond to a sample tested with a single pulse at 740 A/mm^2^, fractographs C and D correspond to a sample tested with multi-pulse current at 45 A/mm^2^, and fractographs E and F correspond to a sample tested with a technical dryer.

No significant differences are observed between the samples tested under electrical current and the one heated with a technical dryer. Upon close examination of the fracture surfaces, typical features of ductile fracture, such as dimples and microdimples, are observed. There are no indications of brittle fracture on the fracture surfaces.

### 3.4. Bending

Regarding bending, Figure 8 displays the stress–strain curves, and Table 3 presents the mechanical characteristics for different test conditions. As shown in Figure 8, the introduction of current or heating by a technical dryer results in a slight decrease in the strain hardening coefficient. Bending without current (curves 1 and 2) exhibits the expected rate dependence of flow stresses, where the stresses increase with higher bending rates.

When a multi-pulse current is applied at a low loading rate, and its density is increased to 30 A/mm^2^ (curves 3 and 4), the sample temperature rises to 220 °C and the flow stresses decrease by approximately twofold.

When the sample is heated with a technical dryer (curve 5), higher flow stresses are observed during bending compared to bending under current at the same test temperature of 100 °C. The curves representing current and technical dryer conditions do not exhibit a smooth transition from the elastic to the plastic region, indicating a change in the strain hardening behavior. The table data also indicate that decreasing the loading rate, introducing current, and heating with a technical dryer result in reduced springback, which is the tendency of the sample to return to its original shape after bending.

The three micrographs in Figure 9, starting from top to bottom, correspond to the bended region at the inner radius (top), the middle of the sample (middle), and the outer radius (bottom). In these micrographs, the etched phase appears white and corresponds to ferrite, while the unetched phase is austenite. The use of the “Beraha” tint etchant (85 mL of water, 15 mL of HCl, and 1 g of K_2_S_2_O_5_) helps distinguish between the two phases.

Upon examination, a slight decrease in the interphase space can be observed when comparing the microstructures at the top and bottom with the one in the middle. This reduction in the interphase space is attributed to the outer regions of the sample experiencing greater deformation due to the bending process.

## 4. Discussion

### 4.1. Tension

When conducting tension tests on duplex stainless steel without current, increasing the strain rate leads to a consistent increase in flow stresses and a decrease in plasticity. This behavior is attributed to the increase in dislocation density and resulting residual stresses [23]. However, the introduction of current has a varied effect on the rate dependence of stress and sample temperature, depending on the specific mode employed.

For tension tests with single-current pulses, increasing the strain rate by an order of magnitude and reducing the pulse frequency result in higher flow stress and lower sample temperature. In the case of multi-pulse current, increasing the strain rate has a minimal impact on the strength characteristics and plasticity, but it contributes to a decrease in temperature. This behavior aligns with the Joule–Lenz law, where a decrease in temperature is observed with shorter residence times at the same current density and pulse duration.

The influence of single-pulse current manifests as downward stress jumps in both the elastic and plastic regions, which are typically associated with the Electric Pulse Effect (EPE) observed in metallic materials. However, since the sample temperature remains relatively low (not exceeding 35–45 °C), the thermal effect of the current is minimal. The decrease in stress jump amplitudes in the elastic and plastic regions with increasing strain rate is likely attributed to machine inertia, an insufficient digitization period, and a high hardening rate, particularly in the elastic region. These results align with literature findings for steel [23,24] (Figure 4, curves 3, 4, 5).

Interestingly, tension tests with single-current pulses exhibit stress jumps not only in the plastic region but also in the elastic region (Figure 4 insert to curve 5). However, the amplitude of stress jumps in the elastic region is significantly lower than in the plastic region. Since elastic deformation does not involve mobile dislocations, the stress jump in this region is primarily caused by thermal expansion during the current pulse. This further supports the existence of the Electric Pulse Effect (EPE) alongside thermal effects and other factors.

In the case of multi-pulse current, stress jumps are not visible on the strain curves, due to the high frequency of pulses. Instead, the primary effect of the current is a significant decrease in flow stresses. The thermal effect becomes more pronounced in this case, resulting in an increase in sample temperature to 100–200 °C. Comparing the effects of multi-pulse current and heating with a technical dryer at the same temperature of 190 °C, it is observed that the decrease in ultimate tensile strength/yield stress is more prominent with multi-pulse current (15%/25%, respectively) than with the thermal effect (10%/20%) (Figure 5, curves 8, 9).

### 4.2. Bending

When comparing the bending stresses between heating with a technical dryer and exposure to current at the same temperature of 100 °C, it is observed that the decrease in stress (∆σ) is much higher in the case of the current. This suggests the presence of an additional contribution and the action of the Electric Pulse Effect (EPE), similar to what is observed in tension tests.

The introduction of pulsed current leads to a decrease in the strain hardening coefficient due to the combined thermal and electroplastic effects of the current [25]. Similar results have been obtained in experiments on the bending of aluminum alloys [26]. This decrease in strain hardening remains consistent across all current modes and temperatures.

The influence of loading rate on hardening is a well-known phenomenon in coarse-grained metallic materials [27]. In general, an increase in deformation rate leads to an increase in the resistance of metals to deformation. This is attributed to the rapid movement of dislocations, resulting in an increased resistance of the crystal lattice to this movement.

By applying the minimum test speed and increased current density, the maximum reduction in springback angle is achieved. One possible explanation for this is the removal of internal stresses that arise during the bending deformation process. This finding is in line with the results obtained in [22].

Overall, the experiments demonstrate the complex interaction between loading rate, current effects, and temperature on the mechanical behavior and springback characteristics of the material. The presence of the Electric Pulse Effect (EPE) highlights the importance of considering additional factors beyond the thermal and mechanical effects in understanding the response of the material during deformation.

### 4.3. Microstructure

No discernible differences in the structure can be observed between the samples subjected to electrical current and those exposed to high temperature without current when examining the fractographs and micrographs at the current magnification levels. This is because the influence of electrical current on dislocation movement is not apparent at these levels of magnification. The effects of electrical current on dislocation behavior and movement are typically observed at higher magnifications or with specialized techniques such as TEM or SEM combined with in situ testing. Microstructural changes induced by electrical current may not be easily noticeable at lower magnifications. Therefore, the absence of visible structural differences in the fractographs and micrographs does not imply that electrical current has no impact; rather, it suggests that the specific effects of electrical current on dislocation movement and microstructure are not evident at the magnification levels employed in this examination.

## 5. Conclusions

The investigation of the influence of various modes and regimes of current, as well as the deformation schemes, on the mechanical behavior and mechanical properties in duplex austenitic–ferritic stainless-steel UNS S32750 (DSS) has shown the following:The application of a pulsed current results in a reduction in the acting stresses, both in tension and bending. This reduction becomes more pronounced as the current density increases and the strain rate decreases. Increasing the strain rate by a factor of ten decreases the contribution of the electroplastic effect from single pulses in reducing flow stresses by 20%. However, the strain rate effect is not observed in the case of a multi-pulse current.The effect of reducing the strength characteristics at the same temperatures in tension or bending is higher for a multi-pulse current compared to external heating. This finding confirms the presence of an electroplastic effect in both deformation scenarios. The relative reduction in flow stresses when using a multi-pulse current is approximately 200 MPa in tension and 800 MPa in bending.In all investigated modes and conditions of current, as well as external heating, the elongation in tension decreases more significantly with higher strain rates and current density.The introduction of a multipulse current during bending leads to a reduction in the strain hardening coefficient and promotes springback.

## Figures and Tables

**Figure 1 materials-16-04119-f001:**
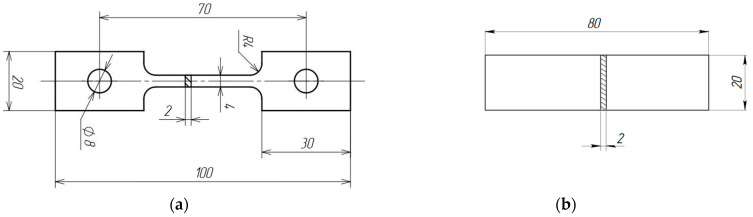
Geometry of the samples for tensile test (**a**) and for bending (**b**).

**Figure 2 materials-16-04119-f002:**
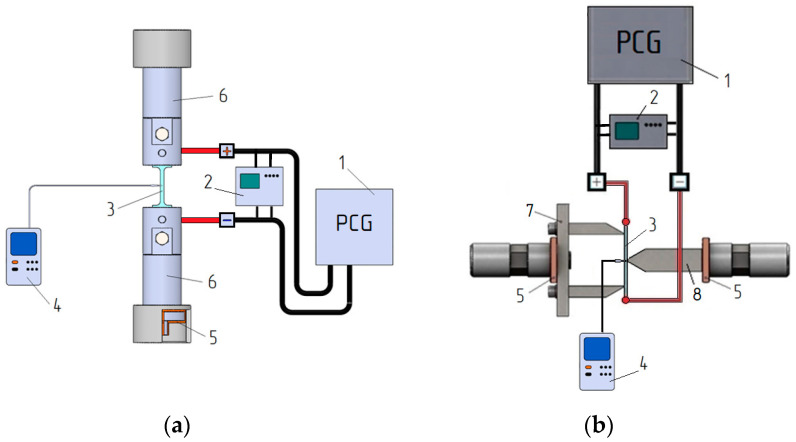
Current supply scheme for tension (**a**) and bending (**b**): 1—pulse current generator; 2—oscilloscope; 3—sample; 4—thermocouple; 5—insulation; 6—clamps of the testing machine; 7—matrix; 8—punch.

**Figure 3 materials-16-04119-f003:**
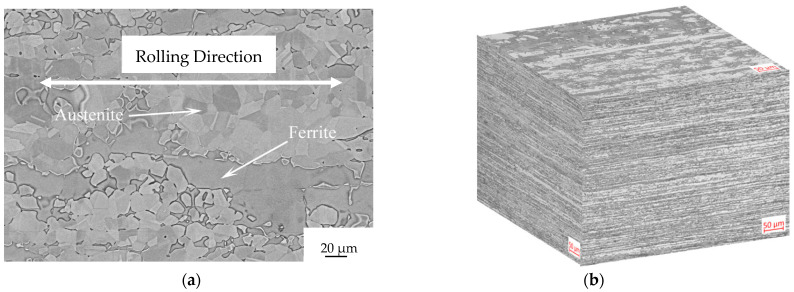
Microstructure of the as-received Steel etched with NaOH at 3 V and 5 s: (**a**) Scanning electron microscope image along the rolling direction and (**b**) optical micrograph along the main three directions.

**Figure 4 materials-16-04119-f004:**
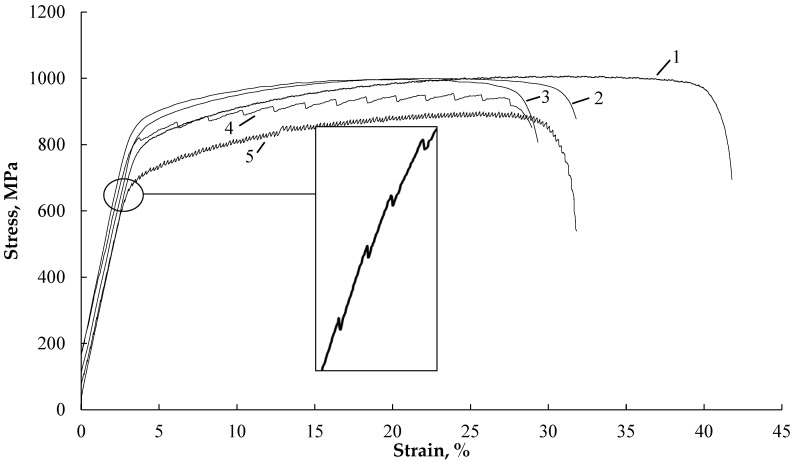
Tensile stress–strain curves: without current: 1—έ—3 × 10^−4^ s^−1^; 2—έ—3 × 10^−2^ s^−1^; single impulses: 3—έ—3 × 10^−2^ s^−1^, j—740 A/mm^2^, 250 μs, T—35 °C, 1 pulse on 5 s; 4—έ—3 × 10^−3^ s^−1^, j—540 A/mm^2^, 1000 μs, T—40 °C, 1 pulse on 5 s; 5—έ—3 × 10^−4^ s^−1^, j—540 A/mm^2^, 1000 μs, T—45 °C, 1 pulse on 2 s. Stress jumps in the elastic region are shown in the inset, curve 5.

**Figure 5 materials-16-04119-f005:**
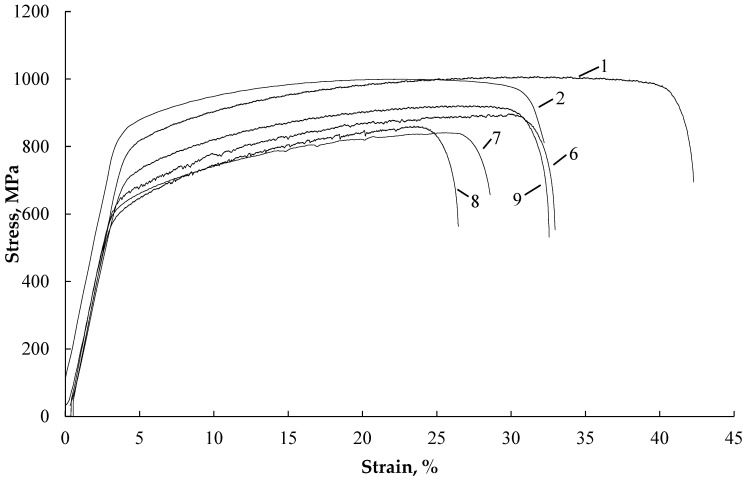
Tensile stress–strain curves: without current: 1—έ—3 × 10^−4^ s^−1^; 2—έ—3 × 10^−2^ s^−1^; multi-pulse current: 6—έ—3 × 10^−4^ s^−1^, j—15 A/mm^2^, 900 μs, T—65 °C; 7—έ—3 × 10^−3^ s^−1^, j—45 A/mm^2^, 100 μs, T—145 °C; 8—έ—3 × 10^−4^ s^−1^, j—45 A/mm^2^, 100 μs, T—190 °C; technical dryer: 9—έ—3 × 10^−4^ s^−1^, T—190 °C.

**Figure 6 materials-16-04119-f006:**
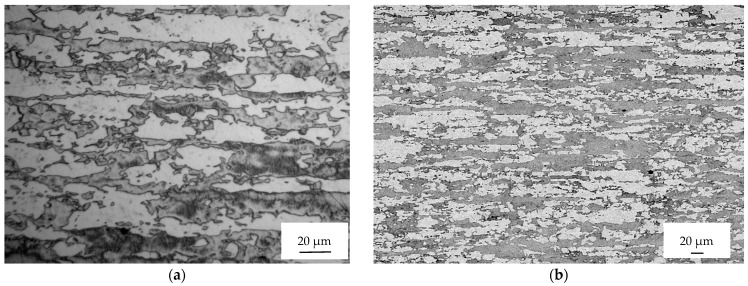
Micrographs of sample 1 (**a**), 3 (**b**), 8 (**c**), and 9 (**d**) far away from the fracture surface.

**Figure 7 materials-16-04119-f007:**
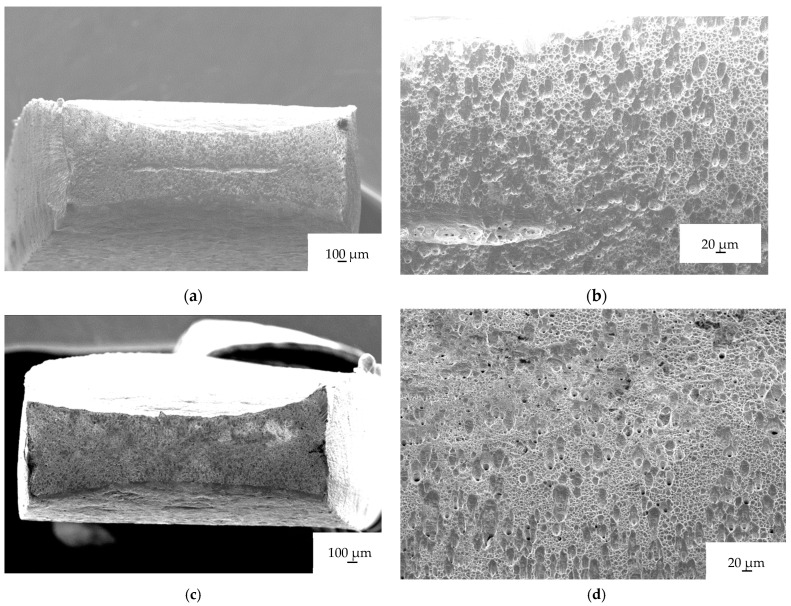
Fractographs of samples 3 (**a**,**b**), 8 (**c**,**d**), and 9 (**e**,**f**). The left side shows the whole fracture surface, while the right side shows a closeup of the fracture surface.

**Figure 8 materials-16-04119-f008:**
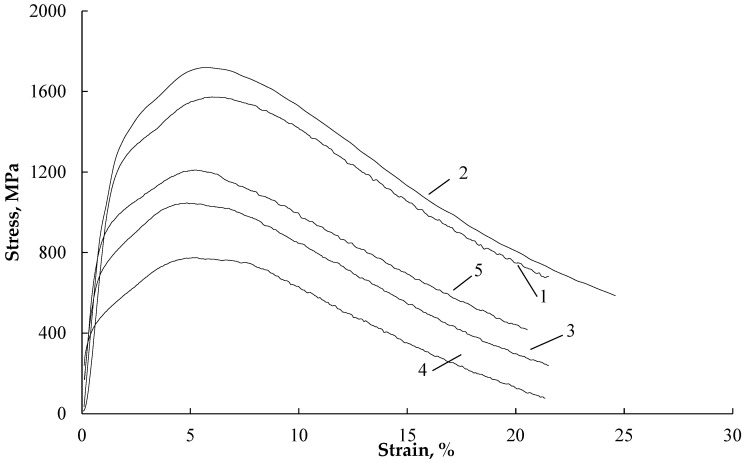
Bending stress–strain curves: 1—without current, 5 mm/min; 2—without current, 200 mm/min; 3—j—20 A/mm^2^ 100 μs, T—100 °C; 5 mm/min; 4—j—30 A/mm^2^ 100 μs, T—220 °C, 5 mm/min; 5—technical dryer T—100 °C, 5 mm/min.

**Figure 9 materials-16-04119-f009:**
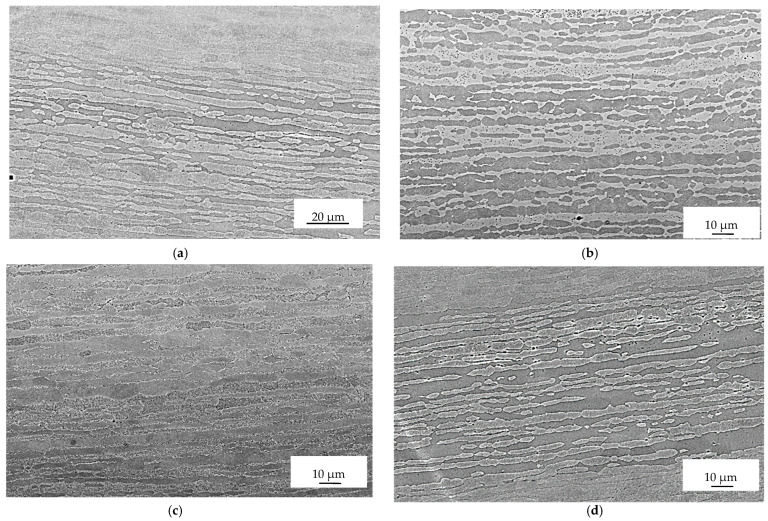
Scanning electron micrograph in backscattered mode of sample deformed in multi-pulse mode with 20 A/mm^2^ (**a**,**c**,**e**) and sample deformed with technical dryer (**b**,**d**,**f**). Inner radius (**a**,**b**), middle (**c**,**d**), and outer radius (**e**,**f**). Etching solution: “Beraha”.

**Table 1 materials-16-04119-t001:** Chemical composition of the investigated DSS (wt.%).

	C	Si	Mn	Cr	Ni	Mo	Cu	W	P	S	N
**UNS S32750**	0.017	0.24	0.88	25.12	6.94	3.85	0.15	—	0.019	0.0010	0.295

**Table 2 materials-16-04119-t002:** Tensile Modes and Mechanical Properties of UNS S32750 Steel.

№	Tension Conditions	Current Regimes			T, °C	Strain Rate, έ, s^−1^	Ultimate Tensile Strength, MPa	Yield Stress, MPa	Elongation, %
		j, A/mm^2^	τ, μs	Frequency, Hz					
1	without current	-	-	-	RT	3 × 10^−4^	1010	765	42
2	without current	-	-	-	RT	3 × 10^−2^	1000	820	32
3	single impulses	740	250	* 0.2	35	3 × 10^−2^	990	805	29
4	single impulses	540	1000	* 0.2	40	3 × 10^−3^	955	785	30
5	single impulses	540	1000	* 0.5	45	3 × 10^−4^	900	665	33
6	multi-pulse	15	900	1000	65	3 × 10^−4^	920	665	33
7	multi-pulse	45	100	1000	145	3 × 10^−3^	840	580	28
8	multi-pulse	45	100	1000	190	3 × 10^−4^	860	570	26
9	technical dryer	-	-	-	190	3 × 10^−4^	895	610	33

*—manual way of changing the frequency.

**Table 3 materials-16-04119-t003:** Bending strength, springback, and bending test conditions.

№	Bending Conditions	Current Regimes			T, °C	Bending Speed, mm/min	Bending Strength, MPa	Spring Angle
		j, A/mm^2^	τ, μs	Frequency, Hz				
1	without current	-	-	-	RT	5	1572	8
2	without current	-	-	-	RT	200	1718	9
3	multi-pulse	20	100	1000	100	5	1042	6.5
4	multi-pulse	30			220		775	6.5
5	technical dryer	-	-	-	100		1208	7

## Data Availability

The data used to confirm the results of this study are found in the article.

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
