# Peer review of "Electroplastic Effect during Tension and Bending in Duplex Stainless Steel"

_materials, 2023, doi:10.3390/ma16114119_

Round 1

Reviewer 1 Report (Previous Reviewer 1)

The authors made significant effort to revise the manuscript, improving the presentation of figures to make them readable. The authors conducted tensile and bending tests to study the electroplastic effect in stainless steels, controlling several testing conditions, including electric pulsing parameters, strain rates, and temperatures. Based on the testing results and microstructure analysis using SEM, the authors drew conclusions about the effects of these parameters on electroplasticity. However, the presentation of data requires further improvement, as some figures appear draft-like. The experimental design is somewhat unclear, as the comparison among testing groups was not fully controlled, given there were multiple variables. In summary, the manuscript requires major revisions before being accepted.

Some detailed comments:

Q1. Did the authors observe deformation twins in austenite grains after the tensile or bending test? If so, any differences in twin destiny might have led to varying working hardening capabilities under different mechanical testing conditions (Line 118, P3).

Q2. Could the authors explain what 'single pulse' and 'multi-pulse' modes are before introducing the testing conditions (Line 136, P4)?

Q3. The serration behaviors in the tensile curves show significant differences (Figure 4). What are the fundamental reasons for these differences, and how do the testing parameters affect the serrations on the stress-strain curves from tensile tests? The authors argue that machine inertia causes the serrations (Line 351, P11). Could the authors explain this in detail?

The manuscript's English needs significant improvement, as many sentences are very difficult to follow.

Author Response

The authors thank and welcome comments from all reviewers.

Reviewer 1

  1. Did the authors observe deformation twins in austenite grains after the tensile or bending test? If so, any differences in twin destiny might have led to varying working hardening capabilities under different mechanical testing conditions (Line 118, P3).

In as-received state the twins are present (fig. 1a). However optical resolution used does not allow concluding on twins influence on strengthening during deformation with current. This point requires a special structural study.

  1. Could the authors explain what 'single pulse' and 'multi-pulse' modes are before introducing the testing conditions (Line 136, P4)?

Impulse current is a short-term surge in electrical voltage in a time interval. We divide the impulse current into two modes: single pulses and multi-pulse current. The differences between these modes lie in the different period T between the current pulses or the different frequency of the pulse current supply.

  1. The serration behaviors in the tensile curves show significant differences (Figure 4). What are the fundamental reasons for these differences, and how do the testing parameters affect the serrations on the stress-strain curves from tensile tests? The authors argue that machine inertia causes the serrations (Line 351, P11). Could the authors explain this in detail?

The recording speed of the machine does not allow to fix data with a short time step, and therefore the jumps are smoothed out and the amplitude and width of the jump are reduced. The stress jump amplitude becomes less than the strain rate.

Comments on the Quality of English Language

The manuscript's English needs significant improvement, as many sentences are very difficult to follow.

English has been greatly improved.

Reviewer 2 Report (Previous Reviewer 4)

The manuscript was well revised, I think that it could be accepted for publication.

Author Response

The authors are grateful for the attentive and favorable attitude to the article.

Reviewer 3 Report (New Reviewer)

The paper entitled "Electroplastic Effect during Tension and Bending in Duplex Stainless Steel" aims to investigate the electroplastic effect in duplex stainless steel under tension and bending. To achieve this objective, the authors conducted a series of experiments to study the relationship between electroplasticity and plastic deformation in duplex stainless steel. Although the study is well-conducted and the results are presented clearly, the paper could benefit from some improvements in terms of English language. Furthermore, in this review, I will provide constructive feedback on some areas that could enhance the quality of the paper, including suggestions for clearer explanations, more precise language, and more concise writing.

1.     The title is first and most important part of your manuscript: “ELECTROPLASTIC EFFECT DURING TENSION AND BEND-1 ING IN DUPLEX STAINLESS STEEL” make it more tentative and just first letter of each word should be capital NOT all.

2.     The Abstract should be written again, its not clear. For example the authors used a lot of data in abstract that they are not necessary. I recommend to reword it again without lot of numeric information. “The relative reduction in flow stresses when using a 15 multi-pulse current is about 200 MPa in tension and 800 MPa in bending” I am as a reader connot understand its improve or its worth??!?!?!? You need to write whole manuscript more clear.

3.     The authors need to bring more references for their facts:” As is known, austenite is a paramagnetic phase, and ferrite is ferromagnetic.”

4.     I recommend drawing a table for the mechanical properties of your selected material. Also, you can mention your as-received material Ultimate tensile strength, Yield stress, and Elongation then readers can compare your materials before/after.

5.     It’s better for authors to provide the company supplying the selected materials. “UNS S32750 duplex stainless steel” is from which company? Also, the annealing process needs to explain in detail. A few minutes can be a big range. Make it more scientifical.

6.     Figure 1, if it’s your original images, you need to put them in results and discussion; if not, you need to bring references, ????? Anyway, it’s not necessary to bring this micrograph here. Even you didn’t explain (a, b). before this figure, you need to introduce your microscopic devices.

7.     Same error: “The microstructure was studied using a Leica DMRE optical 113 microscope. Fracture images were obtained using a scanning electron microscope LEICA 114 Cambridge Stereoscan LEO 440.” Which company/country??? IR-5081/20 horizontal tensile testing ma-133 chine???

8.     Line 169: Two samples were tested for each mode.??? In scientifical experiments at least you need 3 repetitions to calculate the standard deviation.

9.     There are many typography errors inside the manuscript. Read whole the text again. 3*10-4, ×????

10.  Figure 7b, make it similar scale with other images. Also, the images of SEM were taken by high current so it’s a lot of charge in them, if it’s possible make images by 5 kV as current, then you can see better. Also never in the text do you find the etching solution, nor in the sample preparation “Beraha” do you have references?? How do you know its suitable for duplex steel, how about “Adlers” ???

11.  The fracture surfaces should be analysis more and better, diagrams of samples are not clear. For example, it would be good if you explain why and how you have curves like ZIGZAG shapes????

12.  Line 391: “The reason why no differences can be seen in the fractographs and the micrographs 391 of the tensile tested samples and on the micrographs of the bended samples is because the 392 electrical current influences the dislocation movement, which cannot be seen at those 393 magnification levels.” How do you know that??? If it’s fact where are the references????? If it’s your results how you approeached it ???? you brang a lot of micrographs images and finally said there is nothing????!?!?!?

13.  Line 345: “Joule-Lenz law” references? I know it’s W = R * I ² , you need to clarify it? How do you use this law for your discussion part?

14.  Format of the conclusion is not good. You started directly with bullets! I recommend writing one paragraph as an intro and then beginning with bullets. The conclusion must follow your motivation for doing this study.

15.  Check the journal institutions for reference format. Double-check the references. Ref [1] needs to be put in the correct way. Ref [15 and 21] check the characters. Check all Doi again because they are not in the same format.

16.  In the references I didn’t see anyone from 2022-2023??? You strongly need to update your introduction with new research papers. Maybe you need to rewrite the novelty of your study again. Please use the references below in your study:

https://doi.org/10.1016/j.rsurfi.2022.100083

https://doi.org/10.3390/met13020367

Minor editions are needed for the English quality. 

Author Response

The authors thank and welcome the reviewer's comments.

Reviewer 3

  1. The title is first and most important part of your manuscript: “ELECTROPLASTIC EFFECT DURING TENSION AND BEND-1 ING IN DUPLEX STAINLESS STEEL” make it more tentative and just first letter of each word should be capital NOT all.

Fixed.

  1. The Abstract should be written again, its not clear. For example the authors used a lot of data in abstract that they are not necessary. I recommend to reword it again without lot of numeric information. “The relative reduction in flow stresses when using a 15 multi-pulse current is about 200 MPa in tension and 800 MPa in bending” I am as a reader connot understand its improve or its worth??!?!?!? You need to write whole manuscript more clear.

Fixed.

  1. The authors need to bring more references for their facts:” As is known, austenite is a paramagnetic phase, and ferrite is ferromagnetic.”
    [19] Zeng, Y.; Mittnacht, T.; Werner, W.; Du, Y.; Schneider, D.; Nestler, B. Gibbs Energy and Phase-Field Modeling of Ferromagnetic Ferrite (α)→ Paramagnetic Austenite (γ) Transformation in Fe–C Alloys under an External Magnetic Field. Acta Materialia2022225, 117595, doi:10.1016/j.actamat.2021.117595.

[20] Golovin, Yu.I. Magnetoplastic Effects in Solids. Phys. Solid State 200446, 789–824, doi:10.1134/1.1744954.

[21] Okazaki, K.; Kagawa, M.; Conrad, H. An Evaluation of the Contributions of Skin, Pinch and Heating Effects to the Electroplastic Effect in Titatnium. Materials Science and Engineering 198045, 109–116, doi:10.1016/0025-5416(80)90216-5.

  1. I recommend drawing a table for the mechanical properties of your selected material. Also, you can mention your as-received material Ultimate tensile strength, Yield stress, and Elongation then readers can compare your materials before/after.

Table 2 summarizes all the mechanical properties of the as received material.

  1. It’s better for authors to provide the company supplying the selected materials. “UNS S32750 duplex stainless steel” is from which company? Also, the annealing process needs to explain in detail. A few minutes can be a big range. Make it more scientifical.

Added. It is supplied by the Italian division of Outokumpu S.r.l. The annealing process was performed for 10 minutes at 1020 °C.

  1. Figure 1, if it’s your original images, you need to put them in results and discussion; if not, you need to bring references, ????? Anyway, it’s not necessary to bring this micrograph here. Even you didn’t explain (a, b). before this figure, you need to introduce your microscopic devices.

Changed.

  1. Same error: “The microstructure was studied using a Leica DMRE optical 113 microscope. Fracture images were obtained using a scanning electron microscope LEICA 114 Cambridge Stereoscan LEO 440.” Which company/country??? IR-5081/20 horizontal tensile testing ma-133 chine???

Added.

  1. Line 169: Two samples were tested for each mode.??? In scientifical experiments at least you need 3 repetitions to calculate the standard deviation.

We agree with the reviewer's opinion that it is necessary to use 3 samples. And in many of our experiments, so many samples were used that confirmed the reproducibility of the results for duplex steel with a homogeneous structure. In this case, we used only two samples due to insufficient source material, and we consider this sufficient, since the tensile curves were almost the same.

  1. There are many typography errors inside the manuscript. Read whole the text again. 3*10-4, ×????

Fixed.

  1. Figure 7b, make it similar scale with other images. Also, the images of SEM were taken by high current so it’s a lot of charge in them, if it’s possible make images by 5 kV as current, then you can see better. Also never in the text do you find the etching solution, nor in the sample preparation “Beraha” do you have references?? How do you know its suitable for duplex steel, how about “Adlers” ???

Added a link to the article with the chemical composition of Berakhi.

[22] Kinsey, B.; Cullen, G.; Jordan, A.; Mates, S. Investigation of Electroplastic Effect at High Deformation Rates for 304SS and Ti–6Al–4V. CIRP Annals 201362, 279–282, doi:10.1016/j.cirp.2013.03.058.

  1. The fracture surfaces should be analysis more and better, diagrams of samples are not clear. For example, it would be good if you explain why and how you have curves like ZIGZAG shapes????

The study of the direct relationship of tensile curves with single-pulse current and factual images has not been carried out, since our previous long-term experience has shown the absence of such. The physical reason for this is slight heating from single pulses (<50 °C) and reversible stress relaxation.

  1. Line 391: “The reason why no differences can be seen in the fractographs and the micrographs 391 of the tensile tested samples and on the micrographs of the bended samples is because the 392 electrical current influences the dislocation movement, which cannot be seen at those 393 magnification levels.” How do you know that??? If it’s fact where are the references????? If it’s your results how you approeached it ???? you brang a lot of micrographs images and finally said there is nothing????!?!?!?

There are several mechanisms for the influence of impulse current on deformation behavior:

  1. Joule effect, pinch, skin and other similar effects (magnetoplastic)
  2. Electronic wind

There is reason to believe that the first mechanisms in our experiment do not work in tension with single pulses (low temperature and frequency of current, small cross-section of the sample). Therefore, we believe that the main mechanism is the electronic wind. This is consistent with most published studies. Ref. 4  (doi: 10.3390/ma11020185). Unfortunately, there is currently no direct confirmation of this mechanism by structural data.

  1. Line 345: “Joule-Lenz law” references? I know it’s W = R * I ² , you need to clarify it? How do you use this law for your discussion part?

The Joule-Lenz law includes "time", which in our case is related to the strain rate. Therefore, a decrease in the strain rate leads to an increase in the test time and, accordingly, the temperature on the samples. This is what we observe in the experiment.

  1. Format of the conclusion is not good. You started directly with bullets! I recommend writing one paragraph as an intro and then beginning with bullets. The conclusion must follow your motivation for doing this study.

The authors agree with the reviewer's comment. Corrected the text of the conclusion and added a paragraph before the conclusions.

  1. Check the journal institutions for reference format. Double-check the references. Ref [1] needs to be put in the correct way. Ref [15 and 21] check the characters. Check all Doi again because they are not in the same format.

The format of the list of references has been changed.

  1. In the references I didn’t see anyone from 2022-2023??? You strongly need to update your introduction with new research papers. Maybe you need to rewrite the novelty of your study again. Please use the references below in your study: https://doi.org/10.1016/j.rsurfi.2022.100083 https://doi.org/10.3390/met13020367

We agree with your recommendation and have replaced some of the old links with new ones. Thanks for the advice, we will definitely use the links to the links you provided in our future work.

Comments on the Quality of English Language

Minor editions are needed for the English quality. 

The English language has been improved

Reviewer 4 Report (New Reviewer)

1) L. 35: Remove the comma and insert a period.

2) L. 37: Delete the parentheses.

3) L. 38: It should be ...automotive and aviation.

4) L. 43: Keep only refs. [11, 12] because ref. [13] did not investigate the properties of AISI 304L stainless steel.

5) Lines 66 and 67: Please insert a space between the frequency value and its respective unit.

6) In my opinion, the penultimate paragraph of the introduction should have some cohesion with the last. In this sense, I suggest an appropriate combination of both.

7) L. 118: Please use the term "Fig. 1b".

8) L. 134: Certainly if the authors used the equation editor, text would be much more elegant, for example, 3x10-4 s-1 instead of 13*10-4s-1.

9) L. 137: It should be ...500, 550.

10) L. 139: It should be ...1000 Hz; 

11) Please check that the oscilloscope connection shown in Fig. 3 is indeed correct.

12) L. 161: Please insert a space after "(b):".

13) L. 168: It should be "for 10 s".

14) L. 180: Please use the term "Fig. 4".

15) Lines 193-195, 206, 220-231, 241: Please see comment 8.

16) L. 199: It should be "(3, 4, 5)".

17) L. 237: It should be "Table 2".

18) L. 239: It should be "In Fig. 8 and Table 3, are reported, respectively,".

19) Lines 222 and 223: Please insert a space after the commas.

20) L. 235: *Table

21) L. 351: 0.1 s

22) L. 353: 3, 4, 5) [23, 24].

Overall, the text is well written.

Author Response

The authors thank and welcome the reviewer's comments.

Reviewer 4

1) L. 35: Remove the comma and insert a period.

Fixed.

2) L. 37: Delete the parentheses.

Fixed.

3) L. 38: It should be ...automotive and aviation.

Fixed.

4) L. 43: Keep only refs. [11, 12] because ref. [13] did not investigate the properties of AISI 304L stainless steel.

Fixed.

5) Lines 66 and 67: Please insert a space between the frequency value and its respective unit.

Fixed.

6) In my opinion, the penultimate paragraph of the introduction should have some cohesion with the last. In this sense, I suggest an appropriate combination of both.

Fixed.

7) L. 118: Please use the term "Fig. 1b".

Fixed.

8) L. 134: Certainly if the authors used the equation editor, text would be much more elegant, for example, 3x10-4 s-1 instead of 13*10-4s-1.

Fixed.

9) L. 137: It should be ...500, 550.

Fixed.

10) L. 139: It should be ...1000 Hz; 

Fixed.

11) Please check that the oscilloscope connection shown in Fig. 3 is indeed correct.

Fixed.

12) L. 161: Please insert a space after "(b):".

Fixed.

13) L. 168: It should be "for 10 s".

Fixed.

14) L. 180: Please use the term "Fig. 4".

The rules of the journal require writing Figure 4

15) Lines 193-195, 206, 220-231, 241: Please see comment 8.

Fixed.

16) L. 199: It should be "(3, 4, 5)".

Fixed.

17) L. 237: It should be "Table 2".

Fixed.

18) L. 239: It should be "In Fig. 8 and Table 3, are reported, respectively,".

We have replaced this sentence

19) Lines 222 and 223: Please insert a space after the commas.

Fixed.

20) L. 235: *Table

Fixed.

21) L. 351: 0.1 s

Fixed.

22) L. 353: 3, 4, 5) [23, 24].

Fixed.

Round 2

Reviewer 1 Report (Previous Reviewer 1)

I appreciate the effort authors put into addressing the comments and improving the language. Now I suggest the paper be accepted and published.

This manuscript is a resubmission of an earlier submission. The following is a list of the peer review reports and author responses from that submission.

Round 1

Reviewer 1 Report

The authors conducted tensile tests and bending tests with several current pulsing conditions and external heating conditions, using an Austenite stainless steel, to study the electroplastic effect on mechanical properties and deformation modes.

However, the presented mechanical testing results seems not repeatable as the authors only presented one curve for each experiment condition, making the conclusions unreliable. In addition, the manuscript is poorly formated. It only shows fragments in many figures, making them painful to read. For example, the figure is covering the caption of Figure 5. The contents of Figure 6 is missing. Most scale bars in SEM images are not shown correctly. Figure 8 is gone. These issues should be addressed before re-submitting to the journal.

Reviewer 2 Report

The paper seems to have been written in such a hurry that a lot of things had not been done well

Reviewer 3 Report

Pakhomov et al. studied the electroplastic effect during tension and bending duplex stainless steel. The manuscript has certain innovations but there are the following problems that need to be revised.

1 The abstract should give specific quantification parameters.

2 The language of the manuscript needs to be greatly improved.

3 The introduction is unattractive, the author is advised to re-layout.

4 The resolution of the pictures in the manuscript is too low, it is recommended that the author improve it.

5 The Experimental Methods section should be described in more detail.

6 The relevant parameters in Table 2 should be discussed in more detail.

7 The concluding section suggests a more in-depth discussion.

Reviewer 4 Report

This paper deals with the deformation behavior of duplex stainless steel under different modes.

(1) Some introductions on the defcts of the existed modes should be summarized in the introduction part.

(2)Figure format is not correct in some place. some figures are missed in the text. use close format for ally data figure.

(3)The result part and the experiment part should be seperated.

(4)Add some points of the meaning of this work or conculusions in the conclusion part.

(5) Figure 4, why was only one situation posted for the turning point?